# Uranium and Fluoride Accumulation in Vegetable and Cereal Crops: A Review on Current Status and Crop-Wise Differences

**Saloni Sachdeva** [1], **Mike A. Powell** [2], **Girish Nandini** [3], **Hemant Kumar** [3], **Rakesh Kumar** [4]
**and Prafulla Kumar Sahoo** [3,*]

1 Department of Biotechnology, Jaypee Institute of Information Technology, Noida 201309, India
2 Department of Renewable Resources, University of Alberta, Edmonton, AB T5G 2H1, Canada
3 Department of Environmental Science and Technology, Central University of Punjab, V.P.O. Ghudda, Bathinda 151401, India
4 Department of Biosystems Engineering, Auburn University, Auburn, AL 36849, USA
* Correspondence: pk.sahoo@cup.edu.in

**Abstract:** Uranium (U) and fluoride ($F^-$) contamination in agricultural products, especially vegetable and cereal crops, has raised serious concerns about food safety and human health on a global scale. To date, numerous studies have reported U and $F^-$ contamination in vegetable and cereal crops at local scales, but the available information is dispersed, and crop-wise differences are lacking. This paper reviews the current status of knowledge on this subject by compiling relevant published literatures between 1983 and 2023 using databases such as Scopus, PubMed, Medline, ScienceDirect, and Google Scholar. Based on the median values, $F^-$ levels ranged from 0.5 to 177 mg/kg, with higher concentrations in non-leafy vegetables, such as Indian squash "*Praecitrullus fistulosus*" (177 mg/kg) and cucumber "*Cucumis sativus*" (96.25 mg/kg). For leafy vegetables, the maximum levels were recorded in bathua "*Chenopodium album*" (72.01 mg/kg) and mint "*Mentha arvensis*" (44.34 mg/kg), where more than 50% of the vegetable varieties had concentrations of >4 mg/kg. The concentration of U ranged from 0.01 to 17.28 mg/kg; tubers and peels of non-leafy vegetables, particularly radishes "*Raphanus sativus*" (1.15 mg/kg) and cucumber "*Cucumis sativus*" (0.42 mg/kg), contained higher levels. These crops have the potential to form organometallic complexes with U, resulting in more severe threats to human health. For cereal crops (based on median values), the maximum $F^-$ level was found in bajra "*Pennisetum glaucum*" (15.18 mg/kg), followed by chana "*Cicer arietinum*" (7.8 mg/kg) and split green gram "*Vigna mungo*" (4.14 mg/kg), while the maximum accumulation of U was recorded for barley "*Hordeum vulgare*" (2.89 mg/kg), followed by split green gram "*Vigna mungo*" (0.45 mg/kg). There are significant differences in U and $F^-$ concentrations in either crop type based on individual studies or countries. These differences can be explained mainly due to changes in geogenic and anthropogenic factors, thereby making policy decisions related to health and intake difficult at even small spatial scales. Methodologies for comprehensive regional—or larger—policy scales will require further research and should include strategies to restrict crop intake in specified "hot spots".

**Keywords:** agricultural crops; heavy metals; bioaccumulation; agro-ecosystem; non-leafy and root vegetables





## 1. Introduction

The agriculture sector is crucial for economies and societies of countries globally, with substantial implications for both economic development and social welfare [1,2]. Despite the significant value of the global agriculture market (13.4 trillion USD), it is responsible for deteriorating resources (land and water) [2] along with a considerable portion of greenhouse gas emissions—amounting to 30% [3]. Additionally, the agriculture sector controls 10% of the overall 14% of emissions attributed to land use and land cover (LULC) activities. Thus,

it is central to achieving a suite of Sustainable Development Goals (SDGs) agreed to by the United Nations in 2015, ranging from ending hunger (SDG 2), eradicating poverty (SDG 1), promoting gender equality (SDG 5), mitigating climate change (SDG 13), decent work and economic growth (SDG 8), and reducing inequalities (SDG 9) [1,4].

The agricultural sector faces the dual challenge of being both a contributor to and a victim of heavy metal pollution, which primarily originates from sources such as contaminated water, atmospheric deposition, and the use of fertilizers and pesticides [5–8]. Many metals in their ionic form can infiltrate agricultural soils, be absorbed by plants, and enter the food chain, thereby posing health risks to humans upon consumption [9]. The transfer of metals, including arsenic (As), cadmium (Cd), copper (Cu), chromium (Cr), nickel (Ni), manganese (Mn), and lead (Pb), begins with negative impacts on agricultural soil, including crop production, nutrient availability, soil fertility, and ecosystem health [10], and then further extending to accumulation in the food chain and weakening of the immune system, leading to intrauterine growth retardation, impaired cognitive development, and a variety of diseases that impact different bodily systems (such as cancer, dermal problems, respiratory complications, and more) [9,11,12].

Uranium (U) and fluoride ($F^-$) are significant geogenic groundwater contaminants worldwide, yet their accumulation in agricultural systems is often understudied (as illustrated in Figure 1). The use of contaminated water for irrigation, as well as the production and application of phosphoric fertilizers, serve as common anthropogenic pathways for the accumulation of these contaminants in crops [13,14]. Uranium displays dual toxicity, with chemical toxicity prevailing at concentrations below 7–20% as it mimics elements, such as calcium, and interferes with cellular processes [15]; higher concentrations highlight its radioactive properties that can damage DNA, disrupt cell function, and lead to mutations and various radiation-related health problems. In biological fluids, the bioavailable U species (e.g., hexavalent uranyl ions) can be efficiently complexed with biomacromolecules, citrate, and proteins, which are accumulated in bones, kidneys, and the liver [16]. Fluoride is known for its high electronegativity and reactivity. Soluble salts such as sodium fluoride (NaF) are readily absorbed by the bloodstream [17], while protonated fluoride (HF) easily passes through biological membranes [18,19]. In biological systems, $F^-$ has an impact on enzyme activities [20], induces oxidative stress [19], disrupts hormones, and has neurotoxic effects; in addition, the skeleton and teeth are the primary organs where $F^-$ accumulates in the human body and results in skeletal/dental fluorosis [21,22].

The uptake of U by plants is heightened under alkaline pH conditions and increased soil organic matter (SOM) content [23]. Root uptake of U primarily occurs through ion channels that also facilitate the transport of essential elements, such as calcium, iron, and magnesium [24,25]. Anionic forms of U, such as $UO_2(CO_3)_2^{2-}$ and $UO_2PO_4^-$ along with cations (such as $UO_2^{2+}$), may traverse cell membranes via ionic channels, following the essential elements—such as calcium, iron, and magnesium [26]. In the majority of plant species, U is primarily sequestered in the root systems through processes involving complexation, precipitation, and mineralization with phosphate groups [27,28] and polyose of root cell walls [29]. These mechanisms effectively impede U translocation from the plant's roots to the shoots. However, U can be transported via two pathways: (1) through the xylem after forming U chelates, such as $UO_2^-$ citrate and $UO_2^-$ lactate, and (2) through symplastic transport (via tiny openings called plasmodesmata that allow for the direct exchange of molecules), where U ions from the roots transfer to the stele, which is facilitated by transpiration [30]. Exposure to certain levels of U has been extensively documented in the scientific literature to have detrimental effects on various plant physiological and genetic processes, including plant seed germination, growth, photosynthesis, nutrient uptake, and genotoxicity [31,32]. Moreover, U accumulation triggers the excessive generation of reactive oxygen species (ROS) in plants, leading to lipid peroxidation and even cell apoptosis [32,33].

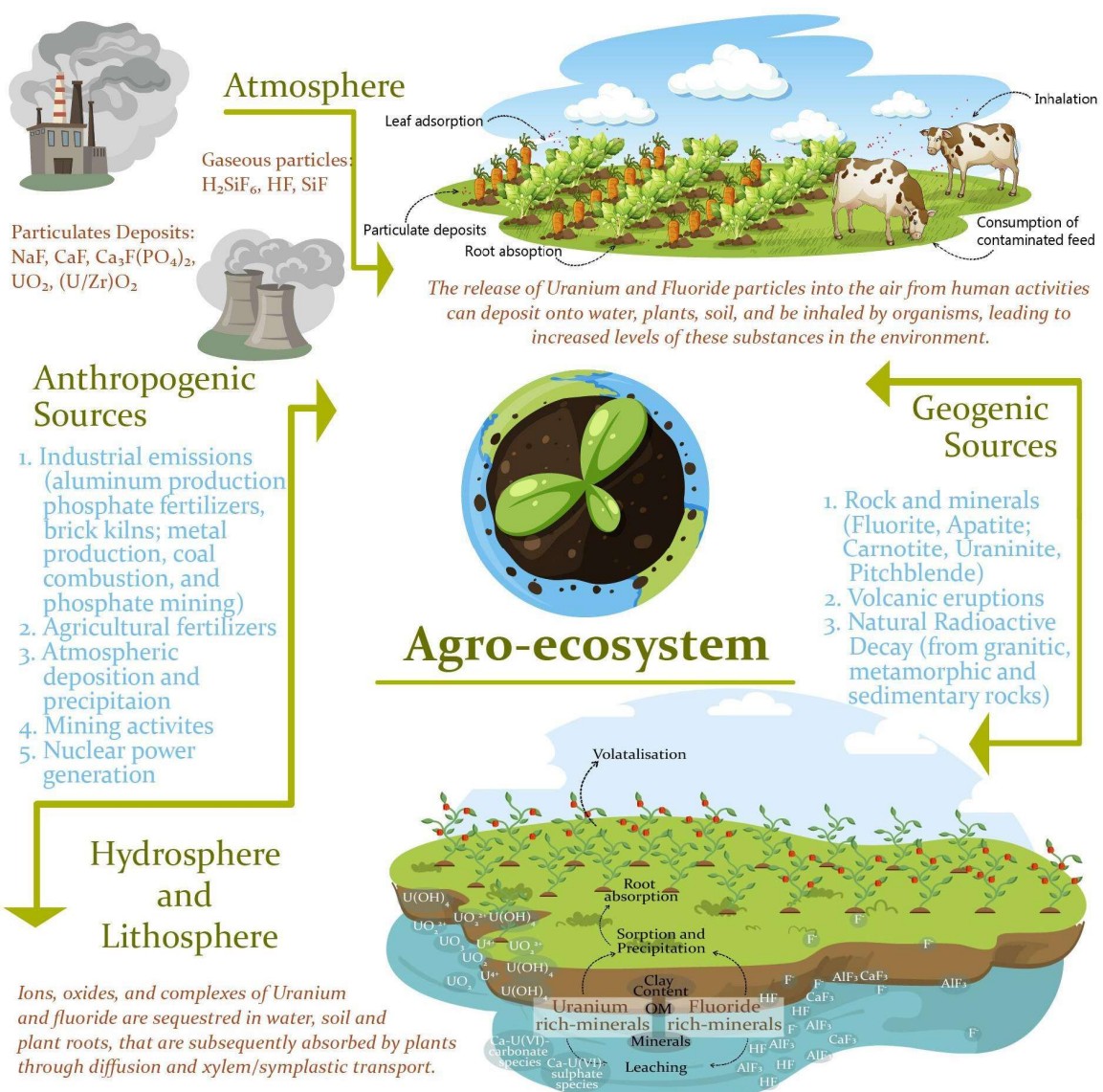

**Figure 1.** Uncovering the pathways of uranium (U) and fluoride (F⁻)from source to agricultural ecosystems. These contaminants can enter agricultural systems from both geogenic and anthropogenic processes and then they can enter the food chain in two main ways: firstly, by being present as particulate matter that humans and animals breathe in, and by foliar uptake by crops; secondly, specific forms of U and F⁻ in water and soil are absorbed by crops through processes, namely, diffusion and xylem/symplastic transport systems.

Fluoride is an electronegative element and its solubility can increase in high total dissolved solids (TDS) water [34], whereas, in soil systems, it is associated with colloidal and clay content with higher levels of Fe/Al hydrous oxides in association with increased solution pH and the ratio of F⁻ to OH⁻ [35]. Moreover, under certain conditions, such as acidity and low soil organic matter, the higher amount of F⁻ leaches into the groundwater [36,37]; in contrast, under alkaline pH conditions, F⁻ exists as free ions available to plants [38]. When plants take in water from the soil, they also absorb F⁻ through their roots, which is transported within the plant through pathways called symplastic transport or xylematic flow, reaching the main transpiring organs in the leaves [39–41]. Furthermore, F⁻ functions as an accumulating toxin in plants, influencing a chain of interconnected metabolic processes, including photosynthesis, respiration, the metabolism of amino acids and proteins, growth, and germination [22,42]. It exerts its effects by interacting with cell membranes and stromal enzymes, leading to the development of necrotic areas in plants [21].

In accordance with the above-mentioned ill effects of U and F$^-$ in soil–plant–human systems, there has been no thorough review on the evaluation of these contaminants in the major agricultural crops around the world. To fill-in this knowledge gap, the present work aims to (1) provide an updated overview of the accumulation potential of U and F$^-$ in various vegetable and cereal crops, and (2) examine the crop-wise differences of these contaminants with various agricultural crops. The results obtained from this research will provide valuable insights, aiding in making informed decisions regarding risk assessments, regulatory advancements, pollution prevention, resource allocation, and the implementation of effective measures to address this issue.

## 2. Methodology

### 2.1. Data Collection and Processing

To conduct the present review work, an extensive search was performed across multiple databases, including PubMed, Google Scholar, and Science Direct. The search encompassed articles on U and F$^-$ contamination in soils and their uptake by crops, without any limitations on publication date. The search terms used were "uranium or U accumulation", "fluoride or F$^-$ accumulation", "uranium accumulation in vegetables", "uranium accumulation in cereal crops", "fluoride accumulation in vegetables", "fluoride accumulation in cereal crops", "fluoride contamination in the agricultural ecosystem", and "uranium contamination in the agricultural ecosystem".

The initial search yielded 92 studies related to F$^-$ and 683 studies related to U (as listed in Supplementary Table S1). To refine the selection, the inclusion criteria required each article to provide information on U and F$^-$ concentrations (mean mg/kg DW $\pm$ standard deviation) in different vegetables or cereals in a soil–crop ecosystem. Values reported in mg/kg fresh weight were converted to mg/kg dry weight as per Staven et al. [43]. This screening process resulted in a final set of 95 studies pertaining to the chemical toxicity of U and F$^-$ only, including 16 studies regarding the radioactive accumulation of U in crops. Relevant details such as the authors, publication year, country, sample size, and mean concentrations (along with standard deviation) of U and F$^-$ in irrigation water, soil, and crops were extracted from each article for further analysis.

### 2.2. Statistical Analysis

The crop data underwent basic descriptive statistical analysis using Microsoft Excel, which involved calculating the central tendency (mean and median) and measures of dispersion (e.g., standard deviation) of F$^-$ and U in crops (both vegetables and cereals), agricultural soil, and irrigation water across multiple countries. The systematic application of these fundamental statistical methods allowed for comprehensive and rigorous dataset exploration, facilitating the derivation of valuable insights and evidence-based conclusions.

### 2.3. Basic Summary of the Compiled Data

In this analysis, the dataset comprised 95 studies, with 1710 records for F$^-$ and 407 records for U in agricultural setup from 1983 to 2023, while vegetables (57.29%) received the most attention in reported studies relative to grains (43.16%) (Figure 2). The countries with the highest number of U investigations were China (6), Portugal (6), Iraq (2), India (2), and Germany (2). For F studies, India had the most research papers (21), followed by China (7), Ethiopia (2), Iran (1), Pakistan (1), Morocco (1), and Kenya (1). However, only a small percentage of the U papers (16.38% and 19.14%, respectively) and F$^-$ papers (83.62% and 80.86%, respectively) provided information on the respective heavy metal concentrations in agricultural soil and irrigation water. The selected studies in this systematic review primarily focused on hot spots and areas known for contaminated soil or water due to anthropogenic activities and/or geogenic factors, leading to a "plateau" effect on graphs and precluding the calculation of specific descriptive statistical parameters.

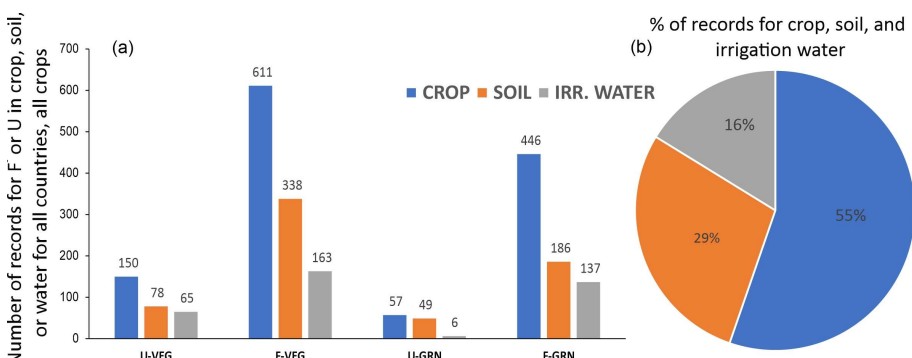

**Figure 2.** (**a**) Number of records for fluoride (F⁻) and uranium (U) in soil–plant system. (**b**) Altogether, the figures for irrigation water amount to 16%, agricultural soil shows 29%, and agricultural crops make up 55%. Note: VEG: vegetable; GRN: grain; IRR.WATER: Irrigation water.

The crops included in the dataset were primarily categorized as vegetables and cereals. Crops within the same family, such as "Green Mustard (*Brassica juncea*)", and "Yellow Mustard (*Brassica nigra*)"; or "Chinese Okra (*Luffa acutangula*)" and "Okra (*Abelmoschus esculentus*)"; or "Green Gram (*Vigna radiata*)" and "Split Green Gram (*Vigna mungo*)"; or "Great Millet (*Sorghum bicolor*)"; "Finger Millet (*Eleusine coracana*)" and "Pearl Millet (*Pennisetum glaucum*)", were not grouped but evaluated individually to assess their bioaccumulation properties. A total of 53 vegetable and 15 cereal crops were classified and named using three-letter crop abbreviations (refer to Supplementary Table S2): ABS—Abyssinian Cabbage; AMR—Amaranth; BCL—Cluster beans; BET—Beet; BKC—Bok Choy; BLS—Black nightshade; BNS—Beans; BPP—Bell pepper; BRN—Brinjal; BTH—Bathua; CBB—Cabbage; CCM—Cucumber; CLF—Cauliflower; CLL—Chilllies; CLR—Celery; CRN—Coriander; CRR—Carrot; DST—Drumstick; FGR—Fenugreek; FNN—Fennel; GBL—Bottle gourd; GBT—Bitter gourd; GPT—Pointed gourd; GRD—Gourd; GRD—Ridge Gourd; INN—Indian Nettle; KAL—Kale; LTT—Lettuce; MBK—Kaali Sarso; MGR—Green Mustard; MNT—Mint; MST—Mustard; MYL—Yellow Mustard; OCH—Chinese Okra; OKR—Okra; ONN—Onion; PBK—Black pepper; PEA—Pea; PMK—Pumpkin; PTT—Potato; RDH—Radish; RDL—Radish Leaves; SBR—Brazilian spinach; SCH—Chinese Spinach; INS—Indian Spinach; SIN—Indian squash; SLL—Slender Leaf; SPN—Spinach; SPT—Sweet Potato; SRR—Sorrel; SWC—Swiss Chard; TMT—Tomato; TNP—Turnip; ARH—Arhar; BBE—Black eye bean; BJR—Bajra; BRY—Barley; CHN—Chana; GGM—Green gram; GGR—Split green gram; KLT—Kulthi; MAZ—Maize; MFN—Finger Millet; MGR—Great Millet; MPR—Pearl millet; RIC—Rice; WHT—Wheat; WIP—West Indian Pea. Regarding the research papers, vegetables were primarily studied, showing significant F⁻ concentrations in spinach "*Spinacea oleracea*" (7.5%), cabbage "*Brassica oleracea var. capitata*" (5.8%), and mint "*Mentha arvensis*" (3.8%), while U studies focused on potato "*Solanum tuberosum*" (16.3%), lettuce "*Lactuca sativa*" (11.5%), carrot "*Daucus carota L.*" (9.6%), beans "*Phaseolus vulgaris*" (9.6%), and radish "*Raphanus sativus*" (8.7%) as the main crops of interest. Cereal crops, including wheat "*Triticum aestivum*", rice "*Oryza sativa*", and barley "*Hordeum vulgare*", were extensively studied, with F⁻ estimation studies comprising 19.4%, 12.6%, and 12.3% of the total, respectively, while U estimation studies primarily focused on rice "*Oryza sativa*" (46.5%) and wheat "*Triticum aestivum*" (39.5%). The WHO has set a maximum permissible limit of 1.5 mg/L for F⁻ [44] and 30 µg/L for U [45] in drinking water. Meanwhile, the permissible limits for agricultural crops are not widely established or standardized. Consecutively, this indicates a lack of awareness about the global translocation and abundance of U and F⁻ in agricultural systems and their detrimental impacts on the food chain.

For vegetables, China reported the largest average F⁻ concentration (1151 mg kg⁻¹) in soil based on only *n* = 4 records, followed by Ethiopia (447 mg kg⁻¹) with *n* = 24, India (114 mg kg⁻¹) with *n* = 279, and Pakistan (8 mg kg⁻¹) with *n* = 27 (Figure 3a). These data fall within the known F⁻ concentrations expected for the respective tectonic setting,

climate, and source rock determinations [46]. Furthermore, the study areas for all countries are situated on soils with diverse geochemical characteristics, ranging from early to late stage magmatic differentiates, metamorphic suites, and thick sediment packages receiving detritus from mountain-building areas (e.g., the Himalayas) to evaporite environments (e.g., the Indo-Gangetic plain). The presence of alluvial plains contributes to the significant spatial variability in $F^-$ concentration [47]. India reported the most results (89%) and exhibited the largest Relative Standard Deviation (RSD) of 146% (Figure 3b), likely indicating the considerable variability in geogenic and anthropogenic sources throughout the country and the presence of "hot spots". In the case of cereal crops, $F^-$ concentrations were reported as an average/RSD in 376 grain crop records (5.7 mg kg$^{-1}$/163%) (Figure 3c), 143 soil records (1796 mg kg$^{-1}$/179%), and 90 water records (3.1 mg kg$^{-1}$/107%). Iran had the highest average $F^-$ concentration in crops (190 mg kg$^{-1}$, $n = 7$), followed by Morocco (6.4 mg kg$^{-1}$, $n = 42$), India (5.2 mg kg$^{-1}$, $n = 312$), and China (3.9 mg kg$^{-1}$, $n = 11$). The substantial contribution from India (83% of total records) significantly influenced the average concentration for all countries and crops. Pearson's correlation for all studies from India revealed a moderate correlation between crop-soil (0.41) and crop-water (0.49) with little difference, suggesting that neither soil nor water significantly impact grain $F^-$ concentrations. However, when considering Singh et al.'s [48] study alone, these correlations become 0.73 for crop-soil and 0.34 for crop-water, indicating that soil has a significantly higher impact on grain $F^-$ concentrations than water.

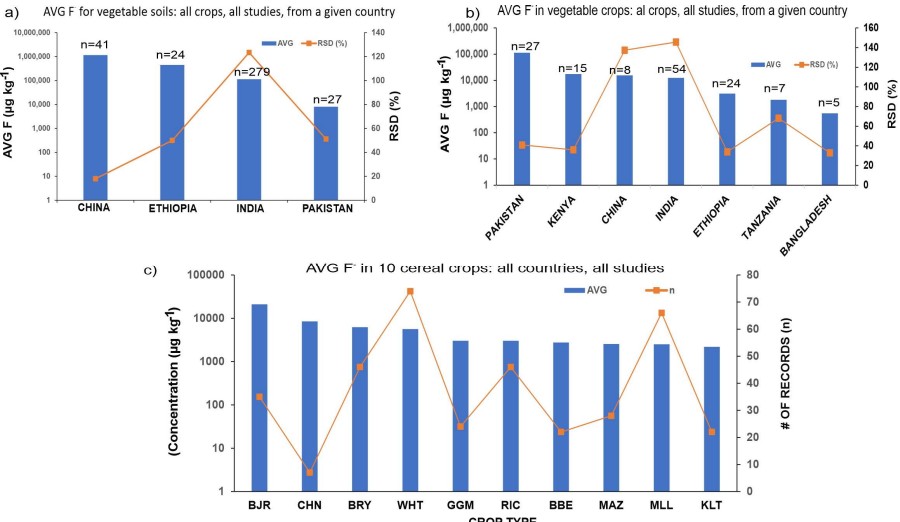

**Figure 3.** Occurrence of fluoride ($F^-$) content in (**a**) soils and (**b**) vegetables across the countries, and (**c**) in cereal crops. Abbreviation: AVG—average; RSD—relative standard deviation; BJR—Bajra; CHN—Chana; BRY—Barley; WHT—Wheat; GGM—Green gram; RIC—Rice; BBE—Black eye bean; MAZ—Maize; MLL—Millet; LKT—Kulthi (Supplementary Table S2).

On the other hand, the average concentration of U in vegetables (4.3 mg kg$^{-1}$/RSD = 73%), soil (89 mg kg$^{-1}$/106%), and irrigation water (0.27 mg L$^{-1}$/46%) from all studies, and all reporting countries, showed significant differences (Figure 4a). The high concentration of U in soils is consistent with its geochemical behavior and known association with soil mineral phases, further supporting the fact that Canadian vegetables have the highest average U concentration (Figure 4b). It is well known that there is a strong correlation between U in soils and plants (or water), depending on the physical parameters of the site and geogenic or anthropogenic sources. India had the next highest U concentration. However, these results indicate that high concentrations of U in soils or plants taken on a regional (country-wise) basis, or from only a few studies that biased their results because of their objective of highlighting health issues rather than "normal" conditions that control metal uptake by plants, may not represent statistically viable results for crop–soil–water interactions. The average U concentration in cereals from all countries and all studies

are (avg./RSD/count): crop (118 µg kg$^{-1}$/140%/38) and soil (2127 µg kg$^{-1}$/80%/38). When cereal crop and soil data are combined in a plot (Figure 5a), a moderately good correlation is observed with an R$^2$ value of 0.61. However, when the four countries are considered separately there are significant differences; in decreasing order of correlation strength: Morocco, R$^2$ = 0.84; Serbia, R$^2$ = −0.5; China, R$^2$ = 0.048; and Iraq, R$^2$ = 0.03. Only Serbia shows a negative relationship. In all studies and for all crops, soil is significantly higher in U than the corresponding cereal crop (Figure 5a–d). Moreover, Morocco (*n* = 1 record) and Serbia (*n* = 3) reported results for MAZ, with soil–crop concentrations of 4600 µg kg$^{-1}$–350 µg kg$^{-1}$, respectively (Figure 5d); China (*n* = 8) and Iraq (*n* = 10) reported records for RIC (soil–crop; Figure 5b) of 4088 µg kg$^{-1}$–338 µg kg$^{-1}$ and 998 µg kg$^{-1}$–8 µg kg$^{-1}$, respectively; Morocco (*n* = 2), Serbia (*n* = 3), and Iraq (*n* = 3) reported results for soil–crop U in WHT of 5000 µg kg$^{-1}$–325 µg kg$^{-1}$, 1610 µg kg$^{-1}$–33 µg kg$^{-1}$, and 1048 µg kg$^{-1}$–14 µg kg$^{-1}$, respectively (Figure 5c). In general, these differences (soil:crop) ranged from 12–124 times for RIC, 13–75 times for WHT, and 13–24 times for MAZ. This instance highlights the potential misinterpretation and unsuitability of averaging large datasets for local/regional policymakers in health/intake management decisions. Hence, more research is required in terms of the estimation of U in agro-ecosystems worldwide.

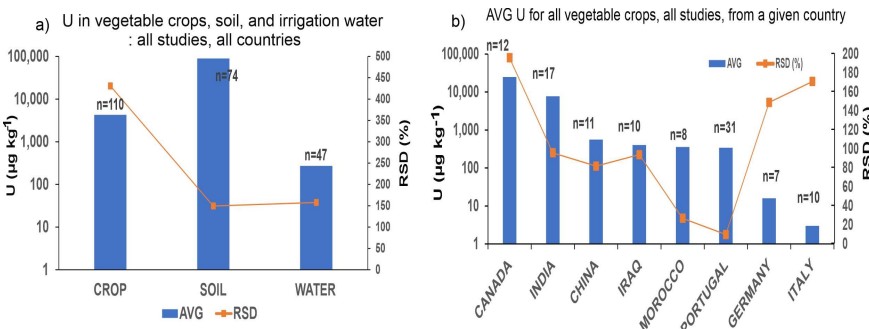

**Figure 4.** Occurrence of uranium (U) in agro-ecosystems with (**a**) the average amount of U found in vegetables, farming soil, and water used for irrigation; (**b**) the average U levels in different countries, along with their corresponding variability (represented as RSD% values). Abbreviation: AVG—average; RSD—relative standard deviation.

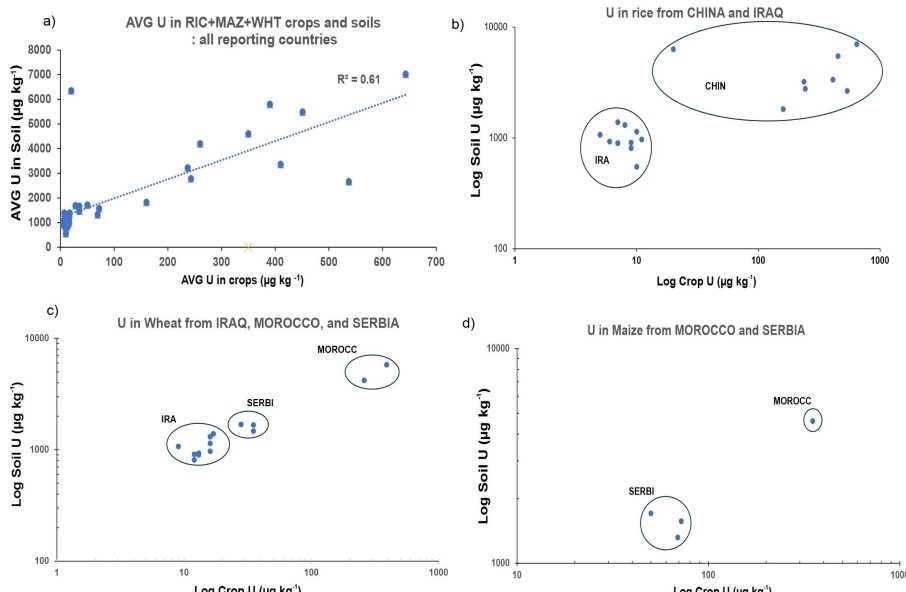

**Figure 5.** The relationship between the average U content in the soil and cereal crops for (**a**) all reporting countries and (**b–d**) country-wise cereals. Abbreviation: AVG—average; RIC—Rice; MAZ—Maize; WHT—Wheat.

## 3. Results and Discussion

### 3.1. Occurrence of Uranium in Soil–Plant System

Uranium, being a radiotoxic and chemotoxic element, poses a higher risk of chemical toxicity due to its long half-life. In the present review, 32 research papers were reviewed, of which 57.14% of studies employed chemical digestion followed by the ICP method for U estimation, 19.04% utilized the fluorescence method, and 23% of the studies employed the CR-39 detector and spectrophotometric methods to determine U accumulation in agricultural ecosystems.

The descriptive statistics of U content in vegetables show a wide variation with the following pattern (Table 1; and detailed information in Supplementary Table S1): RDH (0.013–14 mg/kg; $n = 14$), SPT (0.76 mg/kg; $n = 1$), TNP (0.038–17.95 mg/kg; $n = 4$), CCM (0.0085–0.56 mg/kg; $n = 3$), BET (0.19-0.37 mg/kg; $n = 3$), CLF (0.003–12.46 mg/kg; $n = 3$), LTT (0.008–5.37 mg/kg; $n = 20$), SPN (0.13–0.497; $n = 3$), ONN (0.11–0.279 mg/kg; $n = 6$), and BPP (0.16 mg/kg; $n = 1$), while vegetables such as GBT, PMK, CLL, GBL, and BRN, showed accumulation lower than 0.02 mg/kg. Among cereals, the trend is as follows: BRY (0.3–5.48 mg/kg; $n = 2$), GGR (0.45 mg/kg, $n = 1$), MAZ (0.004–9.69 mg/kg; $n = 9$), and WHT (0.0012–3.01 mg/kg, $n = 23$), while the median concentration of RIC was below 0.02 mg/kg. The study did not incorporate the 17.28 mg/kg accumulation in MGR in its statistical analysis because its main emphasis was on studying hyperaccumulator plants [49]. Nevertheless, the research adds to our understanding by showing that Indian mustard, when grown in calcareous soils, is attributable to the presence of highly soluble uranyl carbonate complexes, which are potentially important for generating rational policy mandates. In the taxonomic context of crops, U concentration in *Lactuca sativa* demonstrated notable variations, with the Romana variety exhibiting higher accumulation and retention in the roots, while the Marady variety showed increased accumulation in the leaves [50]. When comparing the recommended limit of 11.7 ppm set by UNSCEAR (United Nations Scientific Committee on the Effects of Atomic Radiation), the regions of Canada and Portugal exhibited the highest levels of U accumulation, particularly in RDH and LTT, with concentrations ranging from 3 to 14 mg/kg [51–53].

On the other hand, there is a contrasting effect of U in terms of its radioactivity concentration on agricultural crops (Table 2 and detailed information in Supplementary Table S3) leafy vegetable namely MGR (21–27 Bq/kg dry weight), SPN (13.6 Bq/kg dry weight), and SWC (13.54 Bq/kg dry weight) showed a higher accumulation followed by cereal crop MAZ (0.3–30 Bq/kg). A study by Chen et al. [54] revealed that leguminous crops, specifically lupine (83–118) and chickpeas (58–68), accumulated more of a radioactive element (measured in Bq/kg dry weight) compared to leafy vegetables. Chickpeas, in particular, had the highest root-to-shoot ratio at 15.3, indicating a greater accumulation of this element in the edible part of the plant. This finding raises concerns about the potential threat to people living in the area who consume chickpeas regularly. In terms of green leafy vegetables, U radioactivity concentration is more pronounced in the roots of the greens [55,56]; however, SWC stands out with a particularly high translocation coefficient of $5.5 \times 10^{-3}$ [57]. Among non-leafy vegetables, such as PTT and BRN [58], they also demonstrate notably high translocation coefficients, while BET had an accumulation up to 2.98 Bq/kg [59], and CCM showed higher accumulation with median value of 0.42 Bq/kg [60]. Furthermore, U is mainly stored in the cell walls and vacuoles (soluble parts) of the plant [61]. This can alter the plant's usual method of absorbing nutrients in the roots and has the potential to harm both the structure of photosystem II and the electron transport chain in the above-ground part of the plant.

Based on the estimations, U content in agricultural soil exhibited variation in Canada (2–560 mg/kg), and Portugal (1.9–488 mg/kg), while irrigation water exhibited variation in Portugal (0.985–1140 µg/L). The elevated U levels in these regions, in addition to certain regions of India and Europe, are attributed to the presence of granitic rock formations and U mining sites [50,51,62–64]. Additionally, Anke et al. [64] and Esposito et al. [65] found that U mining waste dumps in Germany and Italy stored significantly higher amounts

of U in crops (especially leafy vegetables) grown nearby, while the use of fertilizers in agricultural fields contributes to the increased bioavailability of U in the soil [66–68]. On the contrary, Stojanović et al. [69] indicate that prolonged use of phosphoric fertilizers doesn't significantly increase U contamination in a statistically significant way. However, they did observe that in soils with pH levels above 6 and rich in organic matter, clay, and dust, the risk of U reaching aquifers around agricultural fields is reduced. This, in turn, lowers the amount of U available to plants, and this factor is a significant consideration when assessing the risk of uranium entering the food chain. Additionally, Choudhury & Goswami [70] indicated foliar uptake as another predominant source of U uptake by plants. Another possible cause of U toxicity that hasn't been explored is the release of U into the North Pacific Ocean from the Fukushima-Daiichi nuclear disaster, in addition to naturally occurring sources [71] As the concentration of U increases, it hinders the process of photosynthesis in plants, leading to a reduction in root length. Furthermore, the functionality of protective enzymes in ROS elimination is considerably diminished, leading to inadequate control over the accumulation of free radicals within plant cells. As a result, membrane lipids undergo peroxidation, ultimately leading to cell death [32]. According to experimental findings by Neves et al. [72,73], Amin et al. [74], Gulati et al. [75] and Saric et al. [76], a significant portion of U (65–96%) accumulates in the outer skin or peels of vegetables, such as onions, tomatoes, potatoes, carrots, radish, red beet, and sugar beet. This highlights the importance of removing the peel to minimize human exposure before consumption. However, it should be noted that feeding these vegetables, including their peel, to livestock can introduce U into the human food chain. The negative impact of U contamination is evident through reduced yields of vegetables (up to 87%) reported by Neves and Abreu [77] and Neves et al. [72,77], and reduced yields of cereals (up to 47.7%) reported by Gramms and Voigt [78].

**Table 1.** Descriptive statistics of uranium concentrations, denoted in milligrams per kilogram (mg/kg), showcase heterogeneity across a breadth of agricultural crops.

| Species | Total Count | Min. | Max. | Mean | Median |
|---------|-------------|------|------|------|--------|
| **Leafy Vegetables** | | | | | |
| CBB | 8 | 0.0008 | 17.28 | 2.24 | 0.06 |
| SPN | 3 | 0.13 | 0.497 | 0.28 | 0.20 |
| LTT | 20 | 0.008 | 5.37 | 0.74 | 0.22 |
| CLF | 3 | 0.003 | 12.46 | 4.26 | 0.32 |
| SWC | 1 | 12.22 | 12.22 | 12.22 | 12.22 |
| MGR | 1 | 17.28 | 17.28 | 17.28 | 17.28 |
| **Tuber/Root Vegetables** | | | | | |
| PTT | 20 | 0.0006 | 1.35 | 0.16 | 0.04 |
| CRR | 13 | 0.0008 | 0.67 | 0.19 | 0.06 |
| ONN | 6 | 0.11 | 0.279 | 0.19 | 0.19 |
| BET | 3 | 0.192 | 0.37 | 0.30 | 0.33 |
| TNP | 4 | 0.0038 | 17.95 | 4.81 | 0.65 |
| SPT | 1 | 0.76 | 0.76 | 0.76 | 0.76 |
| RDH | 14 | 0.01324 | 14 | 3.40 | 1.15 |
| **Stem Vegetables** | | | | | |
| GBT | 3 | 0.0044 | 0.49 | 0.17 | 0.004 |
| PMK | 1 | 0.006 | 0.006 | 0.01 | 0.01 |
| CLL | 2 | 0.0044 | 0.008 | 0.01 | 0.01 |
| GBL | 2 | 0.006 | 0.00713 | 0.01 | 0.01 |
| BRN | 4 | 0.0005 | 0.46 | 0.12 | 0.01 |
| OKR | 3 | 0.01723 | 0.536 | 0.19 | 0.02 |
| PEA | 4 | 0.0005 | 0.26 | 0.07 | 0.01 |
| TMT | 14 | 0.0012 | 0.34 | 0.09 | 0.03 |
| BNS | 10 | 0.0081 | 0.645 | 0.19 | 0.03 |
| BPP | 1 | 0.16 | 0.16 | 0.16 | 0.16 |
| CCM | 3 | 0.0085 | 0.56 | 0.33 | 0.42 |

**Table 1.** *Cont.*

| Species | Total Count | Min. | Max. | Mean | Median |
|---|---|---|---|---|---|
| | | **Cereals** | | | |
| RIC | 22 | 0.003 | 0.643 | 0.14 | 0.01 |
| WHT | 23 | 0.0012 | 3.01 | 0.20 | 0.03 |
| MAZ | 9 | 0.0043 | 9.69 | 1.21 | 0.11 |
| GGR | 1 | 0.45 | 0.45 | 0.45 | 0.45 |
| BRY | 2 | 0.3 | 5.48 | 2.89 | 2.89 |

Abbreviations: Min: Minimum; Max: Maximum; CBB—Cabbage; SPN—Spinach; LTT—Lettuce; CLF—Cauliflower; SWC—Swiss Chard; MGR—Green Mustard; PTT—Potato; CRR—Carrot; ONN—Onion; BET—Beet; TNP—Turnip; SPT—Sweet Potato; RDH—Radish; GBT—Bitter Gourd; PMK—Pumpkin; CLL—Chillies; GBL—Bottle Gourd; BRN—Brinjal; OKR—Okra; PEA—Pea; TMT—Tomato; BNS—Beans; BPP—Bell pepper; CCM—Cucumber; RIC—Rice; WHT—Wheat; MAZ—Maize; GGR—Split Green Gram; BRY—Barley (For detailed information, see Supplementary Table S2).

**Table 2.** Uranium radioactivity concentration, denoted in Becquerel Per Kilogram (Bq/kg) showcases heterogeneity across a breadth of agricultural crops.

| Species | Total Count | Min. | Max. | Mean | Median |
|---|---|---|---|---|---|
| | | **Leafy Vegetables** | | | |
| LTT | 9 | 0.013 | 11.2 | 1.86 | 0.110 |
| CBB | 1 | 1.07 | 1.07 | 1.07 | 1.070 |
| SCH | 1 | 1.63 | 1.63 | 1.63 | 1.630 |
| CLR | 1 | 5.55 | 5.55 | 5.55 | 5.550 |
| SRR | 1 | 6.290 | 6.290 | 6.290 | 6.290 |
| INS | 1 | 11.9 | 11.9 | 11.90 | 11.900 |
| SWC | 1 | 13.540 | 13.540 | 13.540 | 13.540 |
| SPN | 1 | 13.600 | 13.600 | 13.600 | 13.600 |
| MGR | 2 | 21 | 27 | 24.00 | 24.000 |
| | | **Root/Stem Vegetables** | | | |
| PMP | 3 | 0.009 | 1.9 | 0.64 | 0.017 |
| OCH | 1 | 0.025 | 0.025 | 0.03 | 0.025 |
| BRN | 3 | 0.028 | 0.032 | 0.03 | 0.029 |
| TMT | 15 | 0.003 | 1.3 | 0.34 | 0.034 |
| GBT | 1 | 0.051 | 0.051 | 0.05 | 0.051 |
| CRR | 3 | 0.05 | 1.26 | 0.49 | 0.154 |
| OKR | 6 | 0.023 | 2.5 | 0.81 | 0.250 |
| INN | 1 | 0.43 | 0.43 | 0.43 | 0.430 |
| DST | 2 | 0.15 | 0.81 | 0.48 | 0.480 |
| CMM | 7 | 0.031 | 6.25 | 1.42 | 0.680 |
| ONN | 5 | 0.012 | 3.12 | 1.07 | 0.900 |
| CLF | 1 | 1.96 | 1.96 | 1.96 | 1.960 |
| BPP | 1 | 2.16 | 2.16 | 2.16 | 2.160 |
| PTT | 3 | 0.005 | 4.72 | 2.32 | 2.240 |
| BNS | 2 | 0.012 | 5.2 | 2.61 | 2.606 |
| BET | 1 | 2.98 | 2.98 | 2.98 | 2.980 |
| | | **Cereals** | | | |
| WHT | 2 | 2.49 | 2.76 | 2.63 | 2.625 |
| BRY | 3 | 3.27 | 4.97 | 4.40 | 4.960 |
| MAZ | 4 | 0.03 | 30 | 14.34 | 13.665 |

Abbreviations: Min: Minimum; Max: Maximum; LTT—Lettuce; CBB—Cabbage; SCH—Chinese Spinach; CLR—Celery; SRR—Sorrel; INS—Indian Spinach; SWC—Swiss Chard; SPN—Spinach; MGR—Green Mustard; PMK—Pumpkin; OCH—Chinese Okra; BRN—Brinjal; TMT—Tomato; GBT—Bitter Gourd; CRR—Carrot; OKR—Okra; INN—Indian Nettle; DST—Drumsticks; CCM—Cucumber; ONN—Onion; CLF—Cauliflower; BPP—Bell pepper; PTT—Potato; BNS—Beans; BET—Beet; WHT—Wheat; BRY—Barley; MAZ—Maize (Supplementary Table S2).

Research findings suggest that leafy crops cultivated in U-contaminated soils exhibit elevated U concentrations [50,65,79–81], while approximately 30% of the absorbed U can be transferred to the edible portions of the plants [72]. In leafy and cereal crops,

U accumulation was most prominent in (decreasing order) the roots/straws, leaves, and edible parts (grains/fruit) [82,83]. Conversely, in non-leafy vegetables, the peels and tubers exhibited a higher accumulation of U (up to 90%) due to their elevated surface area and starch content [78,84–86]. While, Shanthi et al. [87] s demonstrated that edible tubers tend to contain higher levels of radionuclides compared to vegetables. According to the findings of Kadhim et al. [88], the transfer of U from the soil to rice, primarily in the stalks, exhibits a higher rate despite the lower accumulation due to the longer duration of cultivation and the application of water and fertilizers. Notably, purple sweet potato demonstrated the ability to mitigate U through enhancements in energy metabolism, as well as the synthesis of plant hormones (the first messengers) and cyclic nucleotides (cAMP and cGMP, the second messengers) within cells, along with the production of primary and secondary metabolites [86].

In the majority of studies, the bioavailability percentage of U ranged from 1.5% to 17%, which can be attributed to the lower clay content, organic matter, and cation exchange capacity (CEC) in the soil, resulting in a reduced presence of bioavailable U [50,77,89]. A multivariate linear stepwise regression analysis conducted by Ouyang et al. [90,91] observed that the U content in rice grains is predominantly influenced by the levels of U content and pH in the soil. According to Netten and Morley [51], the application of phosphate fertilizers enhances the presence of selenium, phosphates, and carbonates, leading to the precipitation of U on the root nodules, thereby limiting its translocation to the above-ground portions of the crop. Furthermore, research conducted during the summer season has revealed that repeated irrigation water exposure [50,72] along with elevated levels of sulphate and calcium [77] were detected as a result of increased evaporation, which consequently led to increased U exposure and reduced yield in soil–plant systems [50]. According to the findings of Hou et al. [32], the enrichment of U in cucumber and radish can result in the rapid formation of organometallic complexes with U, leading to its accumulation in the human body and causing harmful effects. Also, the germination rate of cucumber and radish in sandy loam soil was maintained, even under high concentration of U stress, indicating its tolerance to U.

Based on highlights of the study Hakonson-Hayes et al. [92], Giri et al. [93] and Hashim & Najam [94], it was found that the main way people are exposed to U toxicological and radiological ill-effects is by directly drinking contaminated groundwater, accounting for more than 85-99% of the health risks for human consumers. Considering established safety measures and relying on oral ingestion, the World Health Organization (WHO) has set a Tolerable Daily Intake (TDI) of 0.6 µg/kg body weight per day for soluble U [95]. The majority of studies indicated TDIs and hazard quotients below 1, suggesting that there are no significant potential risks to human health (such as nephrotoxicity) over a lifetime. Additionally, it was noted that the concentration of U in agricultural crops exceeds 0.04 mg/kg; however, either the bioavailable fraction of U is minimal, or the accumulation predominantly occurs in peels, which can be eliminated, thereby reducing the risk of bioaccumulation in the food chain [72,73,78]. When assessing the overall dietary intake of U as a chemical toxicant, it is notable that in European countries, 59% of the intake is derived from meats and cereals [64,96], while in Morocco, cereals account for 73% of the annual dietary intake [66], potentially resulting in heightened exposure within the food chain.

### 3.2. Occurrence of Fluoride in Soil–Plant System

Fluoride contamination has been relatively neglected in comparison to other organic and inorganic pollutants. For the meta-analysis of $F^-$ accumulation in agricultural ecosystems, 41 research papers were considered. Among these, 81.54% of the studies used the ion-selective electrode method, 11.53% used the potentiometric method, and the remaining 6.93% of the studies utilized the SPADNS, spectrophotometric, and NaOH fusion methods for F estimation.

In terms of F$^-$ content (median values) (listed in Table 3), vegetables showed a wide variation with the following patterns: SIN (158–186 mg/kg, $n = 9$), CCM (9.23–113 mg/kg; $n = 12$), BTH (0.67–98.42 mg/kg; $n = 17$), BCL (61.4–68.5 mg/kg, $n = 9$), MNT (34.5–102.3 mg/kg, $n = 24$), MYL (10.1–24.86 mg/kg, $n = 3$), RDH (4.21–63 mg/kg; $n = 10$), CRN (15–26.94 mg/kg, $n = 4$), OCH (16.8–26.6 mg/kg, $n = 6$), GRD (12.8–25.3 mg/kg, $n = 7$), ONN (3.19–43 mg/kg; $n = 13$), OKR (0.14–75.3 mg/kg, $n = 12$), and KAL (7–29 mg/kg, $n = 11$), and followed by SPN, RDL, GBT, SPT, CRR, PTT, TMT, BRN, BET, MST, LTT, and CLF. Among cereals, the trend is as follows: BJR (1.88–41.04 mg/kg, $n = 36$), CHN (3.26–15.88 mg/kg, $n = 7$), and GGR (0.97–7.3 mg/kg, $n = 2$), while other cereal crops showed accumulation lower than 4 mg/kg. Within the taxonomic context of crops from the families *Amaranthaceae*, *Fabaceae*, and *Poaceae*, the legume species demonstrated a comparatively lower variation in F$^-$ content. In contrast, the *Cucurbitaceae* family exhibited a substantial difference in F levels between cucumber (1.71 mg/kg) and wild cucumber (99.433 mg/kg), while the range of F$^-$ in the *Brassicaceae* family varied from 1.42 mg/kg to 17.87 mg/kg.

Based on the estimations and permissible limit of 1.5 mg/kg for water and 10–50 mg/kg in the soil environment, F$^-$ content in agricultural soil exhibited higher variations in China (1126.6–2450.3 mg/kg), Morocco (1290–14,650 mg/kg), and India (1.3–318 mg/kg). Similarly, irrigation water (majorly, groundwater) displayed F$^-$ concentrations of 7.7 mg/kg in Kenya, 1600 mg/kg in Iran, and 1.82–9973.33 mg/kg in India. The increased F$^-$ levels in these regions can be ascribed to the prolonged accumulation of F from various sources, such as the extensive utilization of phosphate fertilizers; leaching from F-rich rocks, such as apatite, francolite, muscovite, and biotite; industrial activities; and the presence of aluminum plants and brick kilns in close proximity [37,97–99]. The inhalation of F-containing phosphate dust primarily influences the occurrence of endemic human fluorosis [97,100,101], while the existence of dissolved F$^-$ ions, F–metal complexes, and F$^-$ dust particles containing HF, SiF$_4$, and NaF in the soil–plant system leads to the uptake of F$^-$ by plant components, indirectly affecting human health [97,98,102]. Nonetheless, according to Haikel et al. [97] and Zhou et al. [103], it is asserted that washing crops can reduce the F$^-$ content by eliminating the F-containing dust deposited from the atmosphere. Cultivating crops on F-contaminated soil can introduce F$^-$ into the human food chain, including its presence in the forage feed consumed by cattle that graze on contaminated leaves, and Owuor [104] explained that F$^-$ accumulation in crops tends to increase as the crop matures. This widespread issue has a significant global impact, affecting an estimated 200 million people, with China and India being particularly affected—with 60 million and over 66 million individuals affected, respectively. Moreover, the detrimental effects of F$^-$ contamination are evident in decreased yields of cereals (up to 16.1%) and vegetables (up to 43.6%) [100,105,106], posing risks to food security and impeding economic growth.

The highest levels of F$^-$ accumulation were observed in specific regions of India, Kenya, and Pakistan, with 72.5 mg/kg accumulation in maize, 71.2 mg/kg accumulation in green gram, and Bajra showing concentrations ranging from 4.9–41.04 mg/kg, while vegetables recorded an even higher level, such as 158–186 mg/kg in Indian squish, 83.5–113 mg/kg in cucumber, 34.5–102.3 mg/kg in mint, and 0.67–98.42 mg/kg in bathua [99,107,108]. These findings indicate that semi-arid and arid areas with phosphate-rich soils have significant F$^-$ retention properties [109–112] along with air-borne F$^-$ from phosphate fertilizer factories [113]. Moreover, Mustofa et al. [114] and Dagnaw et al. [115] have highlighted that rift valleys, characterized by active tectonic plate movements, volcanic activity, and diverse geological formations, can lead to the enrichment of crops with F$^-$. Several parameters have been identified as contributing factors to increased F$^-$ uptake by plants, such as alkaline pH [38,41,101], reduced SOM, higher calcium content [116], and low clay content [37], as well as foliar deposition [98,102]. For instance, in a recent study, Devi et al. [106] investigated the F$^-$ uptake by different parts of potato plants. The research revealed that the potato roots accumulated F$^-$ in the range of 9.6–121 mg/kg, the shoot accumulated 6–102 mg/kg, and the tuber accumulated 3–79 mg/kg of F$^-$. Additionally, the study found that specific soil conditions, such as a pH of 6.7, electrical conductivity (EC) of 1.76 dS/m,

soil organic matter (SOM) content of 0.98%, and a silt loam texture, supported the mobility and bioavailability of $F^-$. Moreover, higher electrical conductivity (EC) has been shown to regulate root cell membrane permeability [117], along with increased soil sodicity and aluminum silicates, which catalyze the formation of sodium carbonates and sodium bicarbonates, ultimately increasing $F^-$ mobility and uptake by plants [48]. Jha et al. [37] conducted a study comparing three distinct soil types and found that the F accumulation in vegetables and cereal crops followed the order of brick kiln sites > sodic sites > normal sites. This highlights the severe hazards that can be caused to individuals residing in these areas. Additionally, He et al. [118] emphasized that in regions with brick kiln sites, the distribution of $F^-$ dust is primarily influenced by topological and meteorological conditions. Apart from the aforementioned factors, a recent study by Havale et al. [119] revealed that the crop Jowar (sorghum) contains molybdenum, which can retain $F^-$ and reduce copper retention. Similarly, Khandare and Rao [120] emphasized adding molybdenum and aluminum, which reduced $F^-$ uptake by coriander plants. Additionally, the introduction of aluminum reduces $F^-$ uptake by plants by forming the $AlF_3$ complex [102]. Another aspect that has received limited attention is the influence of seasons on $F^-$ accumulation in crops. Singh et al. [48] highlight that the monsoon season leads to increased dissolution of $F^-$ ions in pore water, while conversely, Lakshmi et al. [121] note that during the rabi season, there is limited irrigation water exchange, resulting in the precipitation of fluoride salts in the upper soil layers. Both of these phenomena contribute to an elevated bioaccumulation factor for individuals consuming such food.

**Table 3.** Descriptive statistics of fluoride concentrations, denoted in milligrams per kilogram (mg/kg), showcase heterogeneity across a breadth of agricultural crops.

| Species | Total Count | Min. | Max. | Mean | Median |
|---------|-------------|------|------|------|--------|
| **Leafy Vegetables** | | | | | |
| SCH | 22 | 0.88 | 5.43 | 2.07 | 1.49 |
| SRR | 22 | 1.02 | 4.52 | 2.06 | 1.62 |
| INS | 23 | 0.62 | 6.68 | 2.14 | 1.66 |
| SBR | 22 | 0.96 | 5.79 | 2.40 | 1.91 |
| ABS | 6 | 2.08 | 2.59 | 2.29 | 2.24 |
| SWC | 6 | 2.74 | 5.4 | 3.68 | 3.16 |
| CBB | 30 | 0.054 | 29.8 | 5.79 | 3.30 |
| INN | 21 | 0.58 | 6.95 | 3.46 | 3.48 |
| LTT | 8 | 0.096 | 71.62 | 12.06 | 3.94 |
| MST | 7 | 0.73 | 43.6 | 10.88 | 4.40 |
| CLF | 14 | 1.11 | 78.9 | 11.55 | 4.88 |
| RDL | 9 | 3.21 | 81.2 | 19.95 | 11.94 |
| KAL | 11 | 7 | 29 | 17.27 | 17.00 |
| CRN | 4 | 15 | 26.94 | 20.62 | 20.27 |
| MYL | 3 | 10.1 | 24.86 | 19.65 | 24.00 |
| SPN | 51 | 0.52 | 87.5 | 27.93 | 29.80 |
| MNT | 24 | 34.5 | 102.3 | 51.64 | 44.34 |
| BTH | 17 | 0.67 | 98.42 | 62.68 | 72.01 |
| **Tuber/Root Vegetables** | | | | | |
| TNP | 2 | 0.89 | 1.2 | 1.05 | 1.05 |
| SPT | 4 | 2.2 | 13.43 | 6.54 | 5.27 |
| CRR | 13 | 1.7 | 62 | 18.17 | 5.88 |
| BET | 5 | 0.16 | 20.6 | 8.04 | 9.10 |
| PTT | 21 | 0.96 | 17 | 7.56 | 9.75 |
| ONN | 13 | 3.19 | 43 | 19.32 | 17.61 |
| RDH | 10 | 4.21 | 63 | 27.11 | 21.81 |

**Table 3.** *Cont.*

| Species | Total Count | Min. | Max. | Mean | Median |
|---|---|---|---|---|---|
| **Stem Vegetables** | | | | | |
| GRD | 1 | 0.21 | 0.21 | 0.21 | 0.21 |
| BPP | 8 | 0.27 | 0.77 | 0.52 | 0.50 |
| GPT | 6 | 0.35 | 1.47 | 0.83 | 0.78 |
| BNS | 13 | 1 | 15.26 | 2.98 | 1.47 |
| CLL | 5 | 0.23 | 7.52 | 2.92 | 1.56 |
| GBL | 8 | 1.9 | 14 | 4.79 | 2.30 |
| PMPn | 5 | 2.1 | 3.2 | 2.56 | 2.50 |
| TMT | 24 | 0.12 | 75 | 9.02 | 6.35 |
| BRN | 15 | 1.35 | 75.9 | 13.35 | 7.29 |
| PEA | 3 | 1.6 | 27.1 | 12.35 | 8.34 |
| GBT | 4 | 0.97 | 22.7 | 11.17 | 10.50 |
| OKR | 12 | 0.14 | 75.3 | 18.91 | 17.60 |
| GRD | 7 | 12.8 | 25.3 | 19.06 | 18.50 |
| OCH | 6 | 16.8 | 26.6 | 20.68 | 19.20 |
| BCL | 9 | 61.4 | 68.5 | 64.69 | 64.20 |
| CMM | 12 | 9.23 | 113 | 78.00 | 96.25 |
| SIN | 9 | 158 | 186 | 174.00 | 177.00 |
| **Others** | | | | | |
| BPP | 1 | 0.0009 | 0.0009 | 0.001 | 0.001 |
| AMR | 47 | 0.74 | 7.542 | 2.71 | 2.05 |
| FRG | 3 | 0.94 | 18.24 | 7.09 | 2.10 |
| DST | 23 | 1.45 | 33.139 | 4.31 | 2.30 |
| BKC | 1 | 9.69 | 9.69 | 9.69 | 9.69 |
| CLR | 1 | 10.83 | 10.83 | 10.83 | 10.83 |
| MBK | 18 | 8.96 | 24.86 | 13.53 | 12.49 |
| BLS | 1 | 19 | 19 | 19.00 | 19.00 |
| **Cereals** | | | | | |
| RIC | 74 | 0.07 | 17.44 | 2.56 | 1.19 |
| GGM | 38 | 0.17 | 71.2 | 4.21 | 2.06 |
| MGR | 26 | 0.21 | 2.54 | 1.87 | 2.08 |
| KLT | 22 | 1.3 | 3.44 | 2.19 | 2.20 |
| MAZ | 31 | 0.68 | 72.5 | 5.97 | 2.32 |
| WIP | 21 | 2.06 | 5.09 | 3.03 | 2.60 |
| MFN | 22 | 1.42 | 4.29 | 2.68 | 2.61 |
| BBE | 22 | 1.16 | 4.21 | 2.76 | 2.64 |
| MPR | 23 | 1.51 | 61.3 | 5.34 | 2.76 |
| WHT | 75 | 0.32 | 66.9 | 6.79 | 3.63 |
| ARH | 2 | 2.58 | 4.68 | 3.63 | 3.63 |
| BRY | 45 | 0.9 | 28.4 | 6.25 | 3.65 |
| GGR | 2 | 0.97 | 7.3 | 4.14 | 4.14 |
| CHN | 7 | 3.26 | 15.88 | 8.49 | 7.80 |
| BJR | 36 | 1.88 | 41.04 | 18.77 | 15.18 |

Abbreviations: Min: Minimum; Max: Maximum; SCH—Chinese Spinach; SRR—Sorrel; INS—Indian Spinach; SBR—Brazallian Spinach; ABS—Abyssinian Cabbage; SWC—Swiss Chard; CBB-Cabbage; INN—Indian Nettle; LTT—Lettuce; MST—Mustard; CLF—Cauliflower; RDL—Radish Leaves; KAL—Kale; CRN—Coriander; MYL—Yellow Mustard; SPN—Spinach; MNT—Mint; BTH—Bathua; TNP—Turnip; SPT—Sweet Potato; CRR—Carrot; BET—Beet; PTT—Potato; ONN—Onion; RDH—Radish; GRD—Gourd; BPP—Bell pepper; GPT—Pointed Gourd; BNS—Beans; CLL—Chillies; GBL—Bottle Gourd; PMK—Pumpkin; TMT—Tomato; BRN—Brinjal; PEA—Pea; GBT—Bitter Gourd; OKR—Okra; GRD—Ridge Gourd; OCH—Chinese Okra; BCL—Cluster beans; CCM—Cucumber; SIN—Indian Squash; BPP—Black Pepper; AMR—Amaranth; FGR—Fenugreek; DST—Drumsticks; BKC—Bok Choy; CLR—Celery; MBK—Kali Sarso; BLS—B lack Nightshade; RIC-Rice; GGM—Green Gram; MGR—Green Millet; KLT—Kulthi; MAZ—Maize; WIP—West Indian Peas; MFN—Finger Millet; BBE—Black Eye Bean; MPR—Pearl Millet; WHT—Wheat; ARH-Arhar; BRY—Barley; GGR—Split Green Gram; CHN—Chana; BJR—Bajra(For detailed information, see Supplementary Table S2).

Notwithstanding the higher uptake of $F^-$ by vegetables, the bioaccumulation factor (BCF) generally remained below 1, which was attributed to soil physiochemical proper-

ties, plant-related factors, and the application of gypsum [41,108,118,122]. Enhanced F$^-$ translocation in plants has been suggested to be influenced by higher metabolic rates [123]. Mint exhibited the highest mean BCF values for F$^-$ in brick kiln areas (36.6 mg/kg dwt plant/mg/kg dwt soil) and sodic areas (21.79 mg/kg dwt plant/mg/kg dwt soil), followed by spinach (33.99 mg/kg dwt plant/mg/kg dwt soil) [37]. Other crops such as rice, bitter gourd, wheat, potato, brinjal, tomato, cabbage, and beans also exhibited BCF values greater than 1 [48,124]. In a study by De et al. [125], the investigation of F contamination in agricultural land soil and food crops from endemic regions revealed that non-leafy vegetables had the highest bioaccumulation, followed by leafy vegetables, pulses, and cereals. Cumulative exposure dose (EDI) exhibited a similar trend with non-leafy vegetables having the highest EDI, followed by cereals, pulses, and leafy vegetables. Most studies utilized a reference value of 0.06 ppm/day/body wt for EDI estimation, where some studies [37,105,126,127] reported significantly higher values for cereals, indicating serious implications for the inhabitants of the region.

Based on these observations, several authors have concluded that drinking water contributes significantly (35–47%) to the dietary intake of F$^-$, while cereals and vegetables contribute approximately 25–28% and 25–30%, respectively [123,127–129]. Recently, Mridha et al. [130] reported that the population in Bihar are exposed to higher levels of F$^-$, with water accounting for 95% of the dietary F$^-$ uptake, while the remaining 5% comes from cereals, namely rice (1.37–1.15 mg/kg) and wheat (0.84–0.86 mg/kg). However, considering wheat and rice are staple foods worldwide, Giri and Singh [98] claim that 69% of the risk contribution comes from cereals. These cumulative results highlight the significant source and potential toxic characteristics of F$^-$ content on the overall functioning of the ecosystem. However, the specific mechanisms of F$^-$ uptake and translocation in different plant parts remain unclear, indicating the need for further research in genomics, transcriptomics, and proteomics.

*3.3. Comparison between Vegetable and Grain Crops*

Throughout all the investigated regions, a conspicuous observation emerged, revealing a higher accumulation of heavy metals in vegetables, particularly non-leafy varieties, wherein Indian squash and cucumber demonstrated elevated F$^-$ levels followed by leafy vegetables, such as bathua, mint, spinach, and mustard, while cereals, including barley and split green gram, along with root vegetables, namely radish, turnip, and sweet potato, exhibited heightened U content (Figure 6). This trend contradicts the accumulation of other heavy metals in edible crop parts, as evidenced by Kerketta et al. [131] and Atamaleki et al. [132].

Regarding U studies, data from only four countries were available for analysis, providing usable and comparable records from three cereal crop types (maize, rice, and wheat), while for vegetables, studies were available from eight countries, namely Portugal (3), Iraq (2), India (2), China (2), Canada (2), Morocco (1), Italy (1), and Germany (1), which is not substantial enough to conduct a comprehensive analysis. Vegetables showed higher F$^-$ accumulation compared to cereal food crops; however, the negative consequences were more pronounced for cereals, mainly because of their higher intake [133]. For example, Ranjan and Yasmin [128] demonstrated through experimental analysis that vegetables accumulated F$^-$ in the 3.75–11.820 mg/kg range, whereas cereals contained F in the 1.13–5.58 ppm range. Similarly, the study conducted by Bhargava and Bhardwaj [134] reported F$^-$ content in vegetables ranging from 3.91–29.15 mg/kg, while cereals contained F$^-$ in the range of 0.45–5.98 mg/kg. Furthermore, in leafy vegetables, the accumulation trend was more pronounced in the shoot part, which could be attributed to the strong transpiration pull theory and active water transport in the foliage leaf, as Mondal and Gupta [122] suggested. Conversely, in non-leafy vegetables and cereals, the accumulation of F$^-$ was found to be higher in the root part compared to the shoot part, indicating a mechanism of F ion partitioning, lower permeability through the endodermis, and dilution of F ions, as reported by Devi et al. [106], Battaleb-Looie et al. [38], Li et al. [117], and Jha et al. [135]. Wang et al. [102] suggest that F$^-$, in association with other heavy metals (Ni and Mn), along

with Ca, Mg, and Fe in roots, can form stable complexes with polysaccharides of the cell wall. Additionally, Jha et al. [105] emphasized that F⁻ uptake by wheat is likely to occur through apoplastic rather than symplastic transfer.

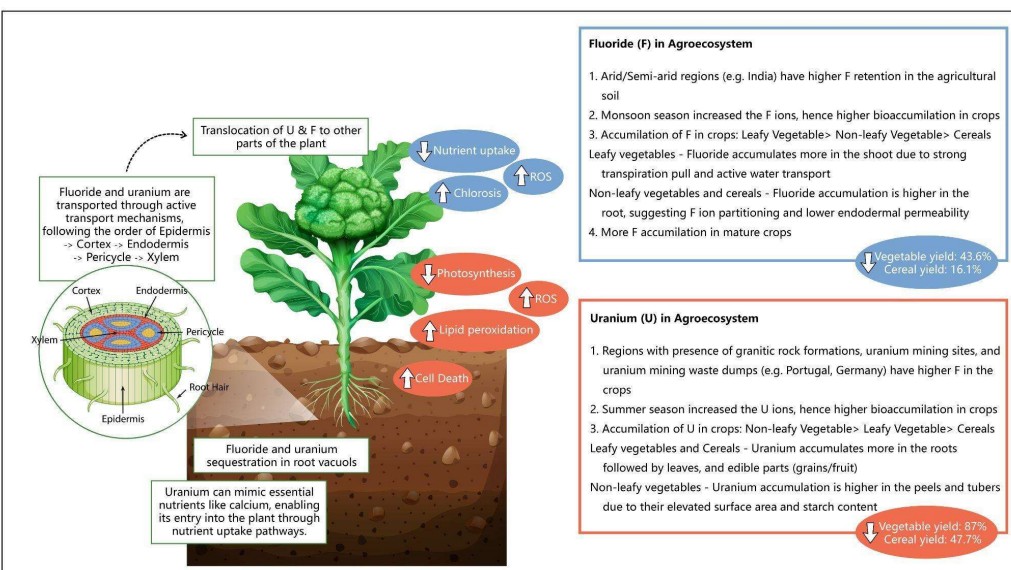

**Figure 6.** Uranium (U) and fluoride (F⁻) uptake in plants: Impacts on yield and physiology. The figure depicts the process of U and F⁻ uptake by plants, involving both active and passive transport systems. Upon entering the plant cells, these toxic elements exert detrimental effects, including heightened cell death, elevated levels of reactive oxygen species (ROS), decreased photosynthesis, and impaired nutrient uptake, as indicated by the blue and orange colors, respectively. These cumulative impacts ultimately lead to a significant reduction in the agricultural yield of cereals and vegetables.

## 4. Limitations

The studies related to U and F⁻ in agro-ecosystems have predominantly concentrated on evaluating their uptake and potential impact on human exposure, while the studies have not encompassed extensive and holistic agricultural investigations on a large scale. Additionally, the ingestion of U in water is not regarded as a significant oral pathway, as indicated by Neves et al. [73] and Wang et al. [14]. However, conducting additional research focusing on alternative pathways including foliar uptake of locally grown agricultural crops, and ingestion of contaminated water and food crops by residents would be valuable, as it has the potential to contribute to elevated U intake levels and, consequently, influence the hazard quotient. The variations in U concentrations and their inconsistent impacts across various parameters, co-existing heavy metals, and soil types highlight the limited understanding of the chemical toxicity of U in agro-ecosystems, thereby impeding a comprehensive assessment of the associated health implications.

Fluoride accumulation exhibits variability within districts of the same region, indicating the need to consider additional soil parameters for a more comprehensive understanding of the phenomenon. Moreover, additional research is required to investigate the impact of soil components on the mobility of F⁻, the involvement of antioxidant enzymes in detoxification processes, the influence of microflora on bioavailability and F⁻ uptake, and the role of various factors (such as soil components, co-existing pollutants, plant exudates, and soil microbial flora) that are yet to be fully understood. This further knowledge will aid in the improved implementation and remediation of sites contaminated with F⁻.

## 5. Conclusions

The comprehensive datasets from India and China have been instrumental in facilitating a thorough analysis of F⁻ and U contamination in agricultural ecosystems; given

that they are agriculture-driven countries, potentially harmful elemental accumulation adversely impacts the agricultural sector and food chain uptake. Regarding U accumulation in crops in fields, the median amount for agricultural crops follows the trend: barley "*Hordeum vulgare*" (2.89 mg/kg), radish "*Raphanus sativus*" (1.15 mg/kg), followed by sweet potato "*Ipomoea batatas*" (0.76 mg/kg), turnip "*Brassica rapa*" (0.65 mg/kg), split green gram "*Vigna mungo*" (0.45 mg/kg), cucumber "*Cucumis sativus*" (0.42 mg/kg), and cauliflower "*Brassica oleracea var botrystis*" (0.32 mg/kg). Among these crops, non-leafy vegetables, namely cucumber, radish, and sweet potato, are commonly eaten by people inhabiting uranium-rich soil, thus posing a threat. Most importantly, it has been observed that cucumber "*Cucumis sativus*" and radish "*Raphanus sativus*" possess the ability to quickly create organometallic compounds with U, which accumulates in the human body, leading to harmful effects. In the case of $F^-$, the accumulation levels were significantly higher, ranging from 0.054–186 mg/kg in vegetables and the order of vegetables based on median concentration (mg/kg) as follows: Indian squash "*Praecitrullus fistulosus*" (177), cucumber "*Cucumis sativus*" (96.25), bathua "*Chenopodium album*" (72.01), beans "*Phaseolus vulgaris*" (64.20), mint "*Mentha arvensis*" (44.34), mustard "*Brassica compestris*" (24), radish "*Raphanus sativus*" (21.81), and coriander "*Coriandrum sativum*" (20.27). For cereals, the concentrations of $F^-$ ranged from 0.07–72.5 mg/kg with median values higher for bajra "*Pennisetum glaucum*" (15.18 mg/kg), chana "*Cicer arietinum*" (7.8 mg/kg), and split green gram "*Vigna mungo*" (4.14 mg/kg). Furthermore, these crops are regularly eaten in regions with elevated $F^-$ levels, such as India, Morocco, and China, which can pose a greater risk to the residents living in these areas. However, it is crucial to acknowledge a significant limitation in these studies, as they include only a restricted range of dietary sources, which can affect the reliability of the findings. In summary, the studies found that non-leafy vegetables (radish, Indian squash, and cucumber) had higher levels of accumulated U and $F^-$. Leafy vegetables and cereals also showed some accumulation, but the focus on non-leafy vegetables has been limited in research so far. Therefore, this systematic review suggests that researchers should investigate this aspect further to gain a better understanding. Moreover, in the context of U, research should prioritize the examination of its chemical toxicity along with its radiational effects.

**Supplementary Materials:** The following supporting information can be downloaded at: https://www.mdpi.com/article/10.3390/su151813895/s1.

**Author Contributions:** Conceptualization, P.K.S., M.A.P. and S.S.; methodology, H.K. and G.N.; software, M.A.P.; validation, M.A.P. and S.S.; investigation, S.S.; writing—original draft preparation, S.S.; writing—review and editing, S.S., P.K.S., M.A.P. and R.K.; visualization, S.S.; supervision, P.K.S. and M.A.P. All authors have read and agreed to the published version of the manuscript.

**Funding:** This research received no external funding.

**Institutional Review Board Statement:** Not applicable.

**Informed Consent Statement:** Not applicable.

**Data Availability Statement:** Data and materials associated with this manuscript is already provided in the Supplementary files.

**Acknowledgments:** P.K.S. acknowledges DST-SERB Core Research Grant (CRG/2021/002567) for providing support to this work. The authors also thank the DST-FIST lab at the Department of the Environmental Science & Technology of the Central University of Punjab, Ghudda, Bathinda for providing technical support.

**Conflicts of Interest:** The authors declare no conflict of interest.

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
