# Peer review of "Uranium and Fluoride Accumulation in Vegetable and Cereal Crops: A Review on Current Status and Crop-Wise Differences"

_sustainability, doi:10.3390/su151813895_

Round 1

Reviewer 1 Report

The manuscript delves into a critical topic regarding the accumulation of uranium and fluoride in agricultural crops, particularly vegetables and cereals. The authors have presented a comprehensive analysis of the available data and discussed the implications of their findings for human health, agricultural practices, and environmental sustainability. Overall, the manuscript demonstrates a solid foundation and contributes to our understanding of these contaminants' impact on the food supply.

The manuscript presents a clear and organized presentation of the results. The findings related to the concentration of uranium and fluoride in different types of crops is presented in a structured manner, making it easy for readers to follow.

With some minor revisions on the points mentioned in comments for authors, I believe it has the potential to make a valuable addition to the body of knowledge in this field.

Reviewer 2 Report

Based on quality of article entitled “A review on Uranium and Fluoride accumulation in vegetable 2 and cereal crops” I recommend to publish the article in the journal after given minor revision. 

Please add values at the bars of the figure 2.

A methodical framework must have to be added in the section of method.

Reviewer 3 Report

Title: Very good-approved

Abstract: Good

Introduction: Line 50-53, please mention names of contaminating sources,

Line-54-56, all metals are in toxic ...not needed for plants growth.

Line-62-65: from which source Uranium is entering in water and then into plants...mention details

Fig 1: Not clear ..use all black color...good font size

No side effects of U and F are mentioned for plants and human beings if taken more than recommended doses.

Methodology: Ok

Fig.2: SEM is missing ...

Fig. 3: SEM is missing what is meaning of connecting lines which connect two-all bars

Table 1: Names of countries add int it and then compare it part-wise and also country wise

Fig. 5: Unreadable ...revise it in black color

Discussion: needs to compare and give rationale with future perspectives of research

Conclusion: too long. make it precise

References : cross check all references in text and section.

English language is OK

Reviewer 4 Report

Dear authors

After a careful review, it was shown that the authors have conducted a serious research on the problem of Uranium and Fluoride accumulation in vegetable and cereal crops.

In my view, I believe that some changes are necessary before accepting this manuscript for publication.

Kind regards
